# A Novel and Rapid Selective Viral Genome Amplification and Sequencing Method for African Swine Fever Virus

**DOI:** 10.3390/v16111664

**Published:** 2024-10-24

**Authors:** Matthias Licheri, Manon Flore Licheri, Kemal Mehinagic, Nicolas Ruggli, Ronald Dijkman

**Affiliations:** 1Institute for Infectious Diseases, University of Bern, 3001 Bern, Switzerland; 2Graduate School for Cellular and Biomedical Sciences, University of Bern, 3012 Bern, Switzerland; 3Multidisciplinary Center for Infectious Diseases, University of Bern, 3012 Bern, Switzerland; 4Institute of Virology and Immunology IVI, 3147 Mittelhäusern, Switzerland; 5Department of Infectious Diseases and Pathobiology, Vetsuisse Faculty, University of Bern, 3012 Bern, Switzerland; 6European Virus Bioinformatics Center, 07743 Jena, Germany

**Keywords:** African swine fever virus, MDA, whole genome sequencing

## Abstract

African swine fever virus (ASFV) is the etiological agent of African swine fever, a highly contagious hemorrhagic disease affecting both wild boars and domestic pigs with lethality rates up to 100%. Until now, the most effective measure to prevent an outbreak of ASFV was early detection. In this situation, whole genome sequencing (WGS) allows the gathering of detailed information about the identity and epidemiology of the virus. However, due to the large genome size and complex genome ends, WGS is challenging. Current WGS workflows require either elaborate enrichment methods or are based on tiled PCR approaches, which are susceptible to genetic differences between ASFV strains. To overcome this, we developed a novel approach for WGS of ASFV, using the Phi29 DNA polymerase-based multiple displacement amplification in combination with only seven primers. Furthermore, we applied an alkaline-based DNA denaturation step to significantly increase the number of viral reads, which resolves the near-full genome of ASFV. This novel isothermal WGS approach can be used in authorized laboratories for the genomic epidemiological analysis of ASFV outbreaks caused by different genotypes.

## 1. Introduction

African swine fever (ASF) is a highly contagious and highly lethal hemorrhagic disease of wild boars and domestic pigs caused by African swine fever virus (ASFV) [1]. ASFV is the only member of the *Asfarviridae* family and has a double-stranded DNA genome between 170 and 190 kb in size, depending on the strain [1]. In 2007, a highly virulent ASFV genotype II was introduced in Georgia and has since spread further to Europe, Asia, Oceania, and the Caribbean [2,3,4,5,6].

Typically, the classification of ASFV into different genotypes is based on the partial nucleotide sequence of the B646L gene, with which 24 different genotypes have been described [7,8]. However, recent analysis has shown that there is insufficient divergence to discriminate between the 24 genotypes, and a new classification of only six distinct genotypes, based on the amino acid sequence of the entire B646L gene, has been proposed [9]. Nevertheless, using such a small genomic region is not ideal for discriminating between different strains, as most of the mutations that might be phylogenetically relevant occur at other genomic locations [9,10,11]. Therefore, whole genome sequencing (WGS) is better suited for ASFV classification. Currently, most of the publicly available whole genome sequences are from ASFV genotype I and II strains responsible for outbreaks outside the African continent, which in general have a high similarity (>99%) [12]. Unfortunately, there are still genomic gaps in the publicly available whole genome sequences for the other 22 ASFV genotypes from Africa, as well as a lack of knowledge about genomic diversity among arthropod and feral host reservoirs [12].

The genomic gap is partially caused by the fact that WGS of ASFV has been challenging due to the size of the viral genome, even with the currently available metagenomic and tiled PCR approaches. The metagenomic approach is based on the enrichment of viral DNA using ASFV-specific RNA baits followed by short-read sequencing, whereas in the tiled PCR approach, pools of PCR amplicons that cover the entire genome are sequenced [13,14,15]. The metagenomic approach allows the generation of high-quality whole genome assemblies but comes with high costs and relatively long processing times and requires specialized equipment. In contrast, the tiled PCR approach is more cost-effective and results in almost complete coverage; but, due to the large number of primers needed for the generation of amplicons spanning over the entire genome, it is sensitive to mutations inserted at the primer binding sites [14]. Furthermore, this approach is limited to a few genotypes because of the limited number of whole genomes of different ASFV genotypes [13]. Combined, these financial and technical constraints limit the feasibility of increasing the number of sequences from genotypes across different geographical locations.

To overcome these limitations, we developed a novel WGS approach based on the multiple displacement amplification (MDA) properties of the Phi29 DNA polymerase to sequence ASFV samples on the Oxford Nanopore Technology sequencing platform. By using ASFV DNA extracted from EDTA blood samples of experimentally infected pigs as a template for Phi29-based MDA, we demonstrated that, within 4 h and with only 7 ASFV-specific primers, we could recover 87% of the 190 kb long ASFV genome. Furthermore, using an alkaline-based DNA pre-treatment step, we could demonstrate that the overall number of viral reads was significantly increased. This novel, rapid, and straightforward WGS method can potentially be used for the real-time genomic epidemiological analysis of ASFV.

## 2. Materials and Methods

### 2.1. Virus and DNA Extraction

The ASFV viral DNA used in this study was extracted at the BSL-3Ag facility of the Institute of Virology and Immunology (IVI) in Mittelhäusern, Switzerland. The ASFV viral DNA was taken from EDTA blood of specific pathogen-free pigs previously infected with ASFV Georgia 2007 (pigs 85, 87, 88) for immunopathogenesis studies [16]. The animal experiments were performed in compliance with the Swiss animal protection law (TSchG SR 455; TSchV SR 455.1; TVV SR 455.163), were reviewed by the committee on animal experiments of the canton of Bern, and were approved by the cantonal veterinary authority under the licenses BE18/2019 and BE46/2022. ASFV DNA was extracted from EDTA blood using the NucleoSpin Blood kit (Macherey-Nagel, Düren, Germany), according to the manufacturer’s instructions. Following nucleic acid extraction, viral DNA was further processed at BSL-2 in accordance with the Swiss Containment Ordinance (ECOGEN A210161-01 and A192517-01).

### 2.2. Viral Culture and DNA Extraction

Due to the limited availability of ASFV DNA, we used a cell-cultured Vaccinia virus as a surrogate for the initial optimization of the assay. The Vaccinia virus was selected due to its genome structure, size, and replication biology similar to that of ASFV. Furthermore, it was readily available as it is a BSL-2 pathogen. To generate the virus stock with pig genomic DNA as the background, we cultured swine testicle cells (ST, American Type Culture Collection CRL-1746, Manassas, VA, USA) in Minimum Essential Medium containing Glutamax (MEM + Glutamax; Gibco, Gaithersburg, MD, USA) and supplemented with 10% heat-inactivated Fetal Bovine Serum (FBS, PAN Biotech, Aidenbach, Germany), 100 μg/mL streptomycin, and 100 IU/mL penicillin (Gibco). One day prior to infection, we seeded 1 × 10^6^ ST cells per well in a 6-well plate. The cells were inoculated with five serial 10-fold dilutions of Vaccinia virus starting with a multiplicity of infection (MOI) of 1. After 16 h of incubation (37 °C, humidified incubator, 5% CO_2_), the cell supernatant was removed, cells were lysed using a DNA/RNA shield (1×; Zymo Research, Irvine, CA, USA), and DNA was extracted using the Zymo Research Quick-DNA Miniprep Plus Kit (Zymo Research) according to the manufacturer’s instructions, and in accordance with the Swiss containment ordinance (ECOGEN A131226).

### 2.3. Primer Design

To specifically amplify ASFV genomes using the MDA approach, we designed primers in silico using the Selective Whole Genome Amplification (SWGA) command-line tool [17]. ASFV (NC_044942.1) or Vaccinia virus (NC_006998) reference genomes were used as the on-target reference, with the pig chromosomal and mitochondrial DNA as the off-target reference (Sus scrofa Sscrofa11.1 pig mitochondrion NC_000845). This resulted in several primer pools, classified based on their binding distance and frequency for both on-target and off-target genomes. We selected six primers for ASFV (ASFV01-06, Appendix A) and Vaccinia virus (VV01-06, Appendix A) that do not bind to the swine mitochondrial DNA (NC_000845) and synthesized them with phosphorothioate bonds at the two 3′ terminal nucleotides in order to prevent exonuclease degradation (Microsynth AG, Balgach, Switzerland).

### 2.4. Sample Pre-Treatment

To improve Phi29-based MDA, we investigated the influence of a DNA pre-treatment step—either of heat denaturation or incubation in an alkaline lysis buffer—compared to untreated DNA. The heat denaturation consisted of incubating DNA in a 7.5 μL reaction containing 1× EquiPhi29 reaction buffer (Thermo Fisher Scientific, Waltham, MA, USA) and 3.5 μM of pooled SWGA primers for a denaturation step (3 min, 95 °C), followed by quenching on ice for 5 min. For the alkaline denaturation approach, we incubated the extracted DNA with an alkaline lysis buffer (pH 14, 0.4 M KOH, 10 mM EDTA, 100 mM DTT) in a sample-to-buffer ratio of 0.85 [18]. After 5 min of incubation at 4 °C, the mixture was neutralized by adding an equal volume of 1 M Tris-HCl (pH 4, Promega, Madison, WI, USA). Pre-treated samples were used thereafter for whole genome amplification based on the MDA.

### 2.5. Multiple Displacement Amplification

A total volume of 2.8 µL native ASFV or Vaccinia virus DNA, or 9.4 µL denatured ASFV DNA treated using either method described above, was used in a 25 µL reaction mixture composed of 1× EquiPhi29 reaction buffer (Thermo Fisher Scientific), 0.6 U/μL EquiPhi29 DNA polymerase (Thermo Fisher Scientific), 1 mM dNTPs (Promega), 1 mM DTT (Thermo Fisher Scientific), and 3.5 μM of pooled SWGA primers (ASFV-ter primer 1/10 of the amount). The two-step incubation included an amplification step for different times (i.e., 1, 2, 4, 8, or 16 h) and temperatures (i.e., 30 °C or 42 °C) followed by an enzyme inactivation step (10 min, 65 °C). The MDA product was purified using the E.Z.N.A. Cycle Pure Kit (Omega Bio-tek, Norcross, GA, USA) according to the manufacturer’s instructions.

### 2.6. Quantitative PCR

To quantify the amount of amplified ASFV DNA, we performed a qPCR using previously published primers and probes specific for ASFV, where the Cy5 fluorescent dye was replaced with FAM [19]. To assess the Vaccinia virus genome amplification yield in the Phi29-based assays, we used previously published qPCR primers targeting the E9L gene [20]. Amplified samples were diluted either 1:100 (ASFV) or 1:10 (Vaccinia virus), from which 2 μL was used in a 10 μL reaction mixture based on the Luna Universal Probe qPCR Master Mix (New England BioLabs [NEB], Ipswich, MA, USA) and according to the manufacturer’s instructions. The analysis was conducted on a QuantStudio 7 Flex Real-Time PCR System (Thermo Fisher Scientific) with a cycle profile involving an initial denaturation step (1 min, 95 °C), followed by 45 cycles of denaturation (15 s, 95 °C), annealing, and elongation (30 s, 60 °C), with a measurement of the fluorescence intensity at each cycle.

### 2.7. Sequencing and Analysis

After purification, the MDA product was quantified using the Qubit dsDNA HS Assay Kit (Thermo Fisher Scientific) with a Qubit 3 fluorometer (Thermo Fisher Scientific), and 400 ng of DNA was used for each barcode during library preparation with the Rapid Barcoding Kit (SQK-RBK110.96 or SQK-RBK114.24, Oxford Nanopore Technologies [ONT], Oxford, UK) according to the manufacturer’s protocol. The library was loaded on a MinION flow cell (R9.4.1 (FLO-MIN106) or R10.4.1 (FLO-MIN114), ONT) and sequenced with a MinION device (Mk1B, ONT) with real-time, high-accuracy basecalling (Guppy 5.0.13 or 5.1.13 or Dorado 7.2.13, ONT) and demultiplexing enabled through the MinKNOW software (21.11.7 or 23.11.7, ONT). Other setups of the sequencing runs included the q-score (≥Q9), the detection of mid-read barcodes, and enabling read splitting. Downstream sequence analysis was performed using a custom-made bash script piping minimap2 (2.21-r1071) for the reference-based assembly as well as medaka (1.9.1, ONT) to polish the consensus sequence [21]. Finally, Samtools (1.18) was used to extract the data necessary to generate the plots with the R package ggplot2 (3.4.4) using R (4.3.1) [22].

## 3. Results

To overcome financial and practical constraints for ASFV WGS, we explored the possibility of using the multiple displacement amplification (MDA) properties of the Phi29 DNA polymerase. However, since the MDA reaction is usually combined with random hexamers, this can result in the overamplification of host-derived nucleic acids, such as the circular mitochondrial DNA. Therefore, we used the Selective Whole Genome Sequencing (SWGA) package to design primers that selectively amplify the viral genome and not the host chromosomal and mitochondrial DNA. Due to the limited availability of ASFV DNA, we performed the first evaluation and associated initial optimization steps using Vaccinia virus as a surrogate.

Using the SWGA command-line tool, we designed primers for the Vaccinia virus (Western Reserve) genome, whereby the pig chromosomal and mitochondrial sequences were used as a background to generate primers that would amplify the Vaccina virus genome selectively. This resulted in a set of six primers that, in silico, targeted the Vaccina virus genome several times (Appendix A). Because the terminal region was poorly represented in silico, we manually designed a primer targeting the terminal regions (VV-ter, Appendix A). The combined set of seven primers was then used to determine the optimal reaction time and temperature yielding the highest Vaccina virus DNA amount. For this, the MDA reaction was conducted at either 30 °C or 42 °C for a total duration of 1, 2, 4, 8, or 16 h in three replicates. Using quantitative PCR, we observed that the increase in DNA yield over time was much slower at 30 °C than at 42 °C (Figure 1). Because it is known that long incubation times can increase the amount of non-specific amplification products, we chose to use an amplification time of 4 h at 42 °C for all further experiments, as this would result in the highest overall yield and would reduce the risk of amplifying non-specific DNA products (Figure 1).

After evaluating the optimal conditions for the Phi29 DNA polymerase-based MDA reaction using the Vaccina virus, we used the SWGA command-line tool to generate a primer set that preferentially binds the ASFV genome over the host chromosomal and mitochondrial DNA. From all the generated primers, we selected six primers that, in silico, bind two to three times across the ASFV genome (Appendix A). Using the previously established amplification condition, we performed the MDA on ASFV DNA isolated from EDTA blood samples of pigs that had previously been infected experimentally with ASFV for immunopathogenesis studies. The MDA product was subsequently sequenced on a flow cell from Oxford Nanopore Technology (Figure 2A). This revealed that after sequence read normalization, approximately 13 percent of all the reads mapped to the ASFV genome, and with a mean coverage depth of 100 reads per site, 74 percent of the ASFV genome could be resolved with a 20× coverage (Figure 2B, Table 1). 

Because the 5′ terminal region was not entirely resolved using six primers, we manually designed a primer targeting the 5′ and 3′ termini of ASFV binding nine times on terminal repeat regions on both ends to improve the recovery of the genomic ends (ASFV-ter, Appendix A). In addition, to increase the potential primer accessibility, we also evaluated whether a heat-denaturing or an alkaline-based-denaturing step before MDA could increase the sequence coverage in this region. Following read normalization, the data showed that the heat-denaturation step displayed no significant difference in the number of viral reads generated compared to untreated DNA, whereas the alkaline denaturing approach showed a significant increase compared to the other two approaches (Figure 2C, Table 1). This indicates that alkaline denaturation improves the efficiency of the specific amplification of ASFV DNA.

When we assessed the sequencing coverage for the two pre-treatment methods, we noticed that both the mean coverage depth and the genome coverage at a 20× coverage for the heat-denaturing pre-treatment were similar to those of no pre-treatment (Table 1, Figure 2B); from this, ≥96 percent of the genome could be resolved at a 1× coverage. The most considerable overall improvement was observed in the alkaline-based pre-treatment, which yielded an approximately 8-fold increase in the mean sequencing depth that, at a 1× coverage, could resolve ≥98% of the ASFV genome (Table 1, Figure 2B).

To assess the sensitivity of the developed approach combined with the alkaline denaturation, we used four 10-fold serial dilutions of ASFV DNA from sample EDTA85 and repeated it in three independent replicates. This showed that the mean depth for a 20× coverage was similar up to a Ct-value of 23.1; thereafter, the amplification and sequencing output was markedly reduced (Figure 3, Appendix A). This indicates that the detection limit for the developed approach corresponded to a Ct-value of approximately 23.1.

These results indicate that the Phi29-based MDA approach combined with an alkaline DNA denaturing step is a suitable approach that can rapidly and selectively amplify the ASFV genome of samples with a Ct-value up to 23, which can then be used directly as a sequencing template.

## 4. Discussion

Here, we present a novel WGS approach for ASFV based on the Phi29 DNA polymerase-based MDA principle. We show that using only seven primers combined with a DNA denaturation step can resolve the near-complete ASFV genome from DNA extracted from EDTA blood samples of two experimentally infected pigs. Furthermore, we show that pre-treatment of the DNA prior to MDA increases the overall genome coverage and the percentage of viral reads, which was higher when using the alkaline lysis buffer. This novel WGS method for ASFV can potentially be used as a novel rapid and selective WGS approach to determine the genomic sequence of ASFV.

We show that an alkaline-based DNA denaturing step, prior to the Phi29-based MDA with seven primers, allows the recovery of near-complete ASFV genomes. Although we reach a sequencing coverage of up to 99%, compared to other WGS approaches, the mean coverage at the 5′ terminal region of the genome remains low. This indicates that the current set of seven primers needs to be complemented with additional primers to increase the coverage in this region regardless of having a unique primer for the terminal genomic ends (Appendix A). A further possible solution would be to increase the overall sequencing depth; however, on the Oxford Nanopore Technology sequencing platform, this would limit the number of samples that can be sequenced simultaneously. An alternative solution would be the design of new primer sets using the recently improved SWGA command line [23]. This would require some additional optimization, but combined with the fundamentals described here, it would potentially generate a suitable alternative WGS approach to determine the genomic sequence of ASFV.

Our study shows that, with a limited set of primers, it is possible to obtain near-complete whole genome sequences for ASFV. In contrast, other WGS methods for ASFV include a metagenomic approach, based on the enrichment of ASFV DNA through specific oligonucleotide baits followed by short-read sequencing, and the tiled PCR approach, which uses 64 primer pairs to generate overlapping amplicons covering the entire ASFV genome [13,14,15]. Thus, compared to these reported WGS approaches for ASFV, our method is less laborious and cost-intensive, and due to the multiple binding sites for each primer, it is also likely more robust towards mutations compared to the tiled PCR approach [24,25,26]. However, due to limited material availability and strict import regulations, we can thus far only show that our method works for the Georgia 2007 strain. Nevertheless, when we evaluated, in silico, the primer binding distribution for other ASFV genotypes, for which a whole genome sequence is available (Appendix A), we observed a similar primer binding distribution as in the Georgia 2007 strain, which indicates that the approach described in this study can potentially be used to amplify the genome of different ASFV genotypes selectively.

In conclusion, we developed a novel WGS approach for genotyping ASFV and other DNA viruses. This versatile ASFV sequencing method can be implemented in authorized laboratories equipped to handle and process extracted ASFV DNA.

## Figures and Tables

**Figure 1 viruses-16-01664-f001:**
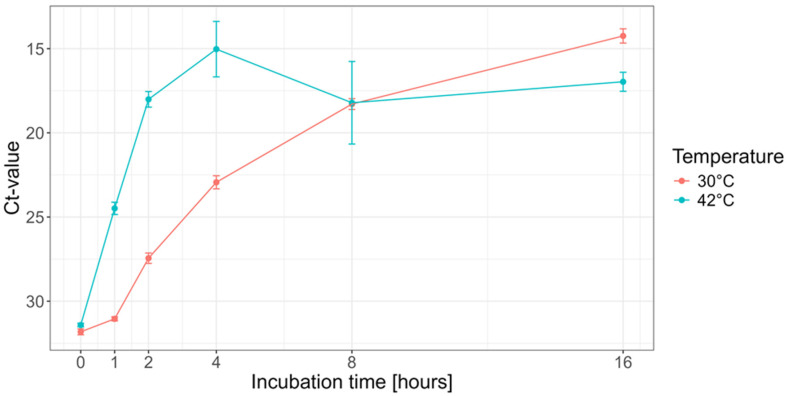
Optimization of the amplification conditions. To optimize the amplification conditions, Vaccinia virus DNA was incubated at either 30 °C or 42 °C for 1, 2, 4, 8, and 16 h, and, after quantification by qPCR, the Ct-value was plotted (*y*-axis). The 0 h represents the viral load used as input for each reaction.

**Figure 2 viruses-16-01664-f002:**
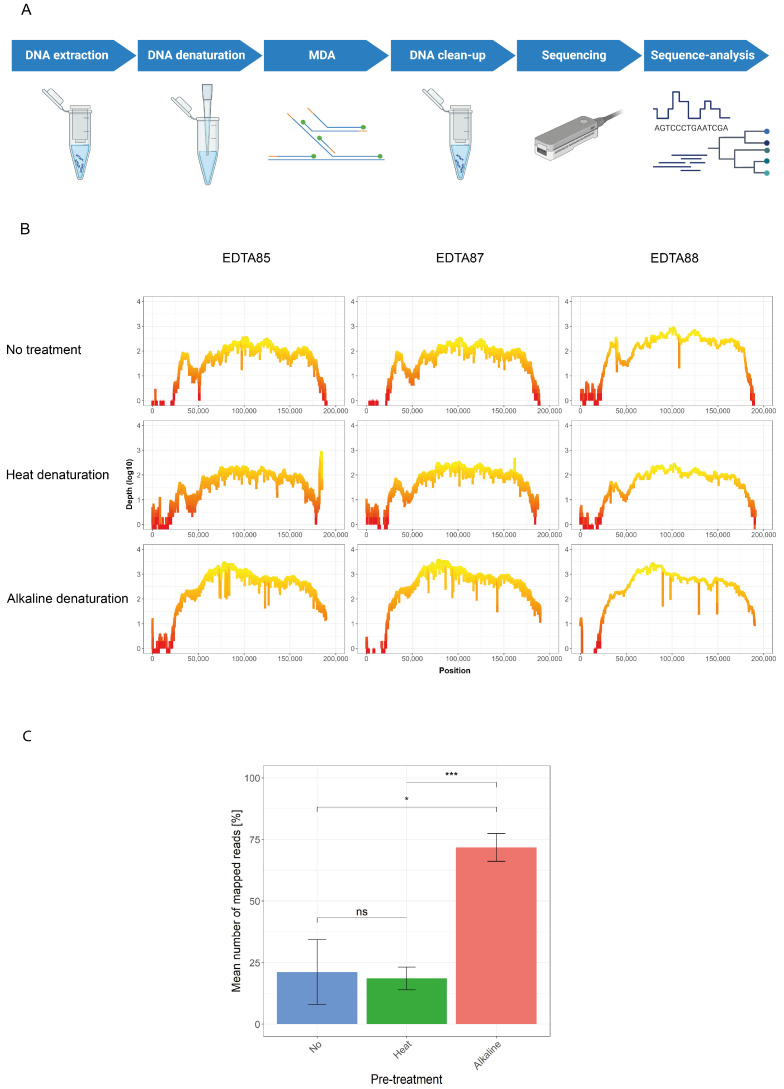
Amplification and sequencing approach for ASFV. (**A**) Schematic overview of workflow established in this study (Created in BioRender. Licheri, M. (2024) BioRender.com/b70m288); (**B**) the sequences obtained with three ASFV samples (EDTA85, 87 and 88) amplified with or without pre-treatment were used to generate coverage plots showing the sequencing depth (log10 scale, *y*-axis) in the function of each nucleotide position (*x*-axis); (**C**) the percentages of the obtained mapped reads were compared between the three different pre-treatment conditions (no, heat, or alkaline pre-treatment; ns: not significant, *: *p* ≤ 0.05, ***: *p* ≤ 0.001, paired *t*-test).

**Figure 3 viruses-16-01664-f003:**
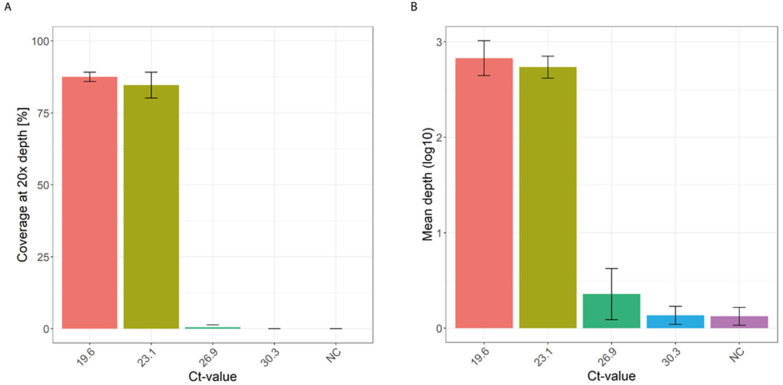
Sensitivity of the SWGA MDA combined with the alkaline denaturation. (**A**) Four 10-fold serial dilutions of ASFV (on the bottom is the corresponding Ct-value, NC = negative control) were amplified in independent replicates using the developed approach, resulting in similar 20× coverage up to a Ct-value of 23.1; (**B**) the corresponding mean sequencing depth of the four 10-fold serial dilutions shows a marked reduction after the second 10-fold serial dilution.

**Table 1 viruses-16-01664-t001:** Comparison of sequencing results based on the different pre-treatment options.

Sample	Ct-Value	Denaturation	Terminal Primer	Mapped Reads	Coverage (1×)	Coverage (20×)	Mean Depth
EDTA85	19.7	No	No	13.7%	89.7%	74.1%	103.9
EDTA87	19.4	No	No	13.3%	89.8%	74.0%	97.1
EDTA88	19.5	No	No	36.4%	99.7%	82.1%	241.3
EDTA85	19.7	Heat	Yes	20.7%	96.7%	72.4%	90.6
EDTA87	19.4	Heat	Yes	21.7%	97.0%	76.1%	119.4
EDTA88	19.5	Heat	Yes	13.3%	95.7%	74.4%	87.0
EDTA85	19.7	Alkaline	Yes	73.1%	98.8%	87.0%	775.3
EDTA87	19.4	Alkaline	Yes	76.6%	91.9%	87.8%	983.6
EDTA88	19.5	Alkaline	Yes	65.5%	92.9%	86.7%	663.3

## Data Availability

The data presented in the study have been deposited in the European Nucleotide Archive (ENA) at EMBL-EBI under the accession number PRJEB75451 (https://www.ebi.ac.uk/ena/browser/view/PRJEB75451 (accessed on 13 October 2024)).

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
