# Peer review of "A Novel and Rapid Selective Viral Genome Amplification and Sequencing Method for African Swine Fever Virus"

_viruses, 2024, doi:10.3390/v16111664_

Round 1

Reviewer 1 Report (Previous Reviewer 1)

Comments and Suggestions for Authors

In this manuscript, Licheri et al, described how they developed and evaluated the use of the Phi29 DNA polymerase-based multiple displacement amplification method in combination with only 7 primers, as a new approach for Whole Genome Sequencing (WGS) of African Swine Fever virus (ASFv), including the application of an alkaline-based DNA denaturation step to increase the number of viral reads. The authors conclude that this novel isothermal WGS can be used for the genomic epidemiological analysis of ASFV.

Although the authors described a new approach to overcome limitations in the actual WGS, there are two principal reasons of why I am rejecting the manuscript: 1)  the number of samples used by the authors to evaluate this methodology,  and 2) Only one genotype was tested.

The number of samples is very limited (only 3 EDTA blood pig samples evaluated): the number of samples must be increased, not only in number, also different genotypes. The authors are using only genotype II (form Georgia strain, not even using other strains from the same genotype). The authors must include different sample types (blood, nasal swabs, oral swabs, tissue homogenates, etc.) to expand and cover possible problems/inhibitions that may arise due the methodology used.

The approach to get better WGS is really good, but without the proper materials to perform the methodology, the work is not complete.

Author Response

We acknowledge the importance of including a greater variety of samples and genotypes to validate the methodology further. However, as outlined in our previous response and reflected in the revised manuscript, we are constrained by national and international regulations that limit our access to other ASFV genotypes. This limitation was explicitly discussed in our manuscript to ensure complete transparency.

Despite these constraints, we made significant efforts to incorporate feedback, including optimizing and refining the methodology where possible. furthermore, we have explained in the discussion section that although only a single genotype was tested, this study show sa proof-of-concept approach. The successful use of Phi29 DNA polymerase-based multiple displacement amplification with 7 primers, combined with an alkaline-based DNA denaturation step, establishes a foundation for future studies involving other genotypes and sample types, or even viruses, as we have we have recently demonstrated DOI: https://doi.org/10.1038/s41598-024-73613-3. Thereby further supports the reliability of our approach, even beyond the scope of ASFV.

While we recognize that expanding the sample types (e.g., blood, nasal swabs, oral swabs, tissue homogenates) would provide additional insights, it is important to note that such expansions would require further time and resources (e.g., animal experimentation, import licenses, etc) that fall beyond the scope of the current work.

Our primary aim in this study is to introduce a novel isothermal Whole Genome Sequencing (WGS) method for ASFV, which we believe can now serve as a valuable tool for genomic epidemiological analysis in future research involving a broader range of genotypes and sample types.

Thank you for your time and consideration.

Reviewer 2 Report (Previous Reviewer 2)

Comments and Suggestions for Authors

 I did read this and think the paper can be accepted now.

Author Response

Thank you for your time and consideration.

Round 2

Reviewer 1 Report (Previous Reviewer 1)

Comments and Suggestions for Authors

The presented restrictions impede the performance of these additional experiments, I decided to accept it.

This manuscript is a resubmission of an earlier submission. The following is a list of the peer review reports and author responses from that submission.

Round 1

Reviewer 1 Report

Comments and Suggestions for Authors

The manuscript "A novel and rapid selective viral genome amplification and sequencing method for African swine fever virus" presented by Licheri M et al., shows important results for improvement in whole genome sequencing. 

In this manuscript, the authors reduced  the number of primers using for sequencing a genome like African swine fever virus, which is a DNA virus difficult to get whole genomes due the size of its genome. The new methodology that the authors described in this manuscript looks like can be use to provide a good quality in the sequencing of this virus. However, I have some comments about it:

The authors are using the Nanopore platform for doing the sequencing, but they do not include any comparison with other methodology, like Illumina sequencing for example, authors should include this data to make a comparison with the data obtained from Nanopore platform.

The authors used blood samples from animals infected with Georgia2007. They mentioned that samples are  from a pathogenesis experiment. It is important to include the dose and route of inoculation used to infect the animals. The authors must include the day post infection when samples were collected, and a titration experiment of these samples.

The authors showed comparison of sequencing results from two different animals ("85 and 87"). Why the authors are not including more samples from more animals infected? The authors must include samples from at least 3 to 5 different animals to have a robust data.

In general, the manuscript contains a relevant methodology to obtain whole sequencing genomes of ASFV, but as they mentioned the methodology has the limitation to only Georgia2007 strain. The authors should include an analysis where they can explain this limitation, with a figure or a table.

Reviewer 2 Report

Comments and Suggestions for Authors

The authors investigated a new method for genome amplification and sequencing of ASF using whole genome sequencing. The current method uses only 7 primers across the genome and gives coverage of up to 99% with reduced time. The latter may allow for more ASF full genomes to be sequenced to address the questions about differences between genotypes and within outbreaks. Although only one ASFV was tested, it could be useful to in future use the method for analysis of more genotypes to see if it could be universally applied. I do not think you need to include more control steps at present, but you could work with reference laboratories to analyse the possibility of using the method on more than just genotypes currently available. Although the authors describe the potential use in conventional laboratory settings, the WOAH requires confirmation from a Reference laboratory. As ASF is a notifiable disease, WOAH will not allow working with potential infectious material in conventional laboratory due to risk of disease spread. I suggest you state just ease of identification and not where it could be used. 

The description of a novel genome amplification and sequencing method for ASF holds merit. Although it was only tested on one ASFV it would be useful to evaluate the protocol on other ASFV. 

Comments on the Quality of English Language

Figure 1 should read "used as input". 

Reviewer 3 Report

Comments and Suggestions for Authors

In this manuscript, Licheri et al, developed and evaluated the use of the Phi29 DNA polymerase-based multiple displacement amplification method in combination with only 7 primers, as a new approach for WGS of ASFV, applying an alkaline-based DNA denaturation step to significantly increase the number of viral reads. The authors conclude that this novel isothermal WGS can resolve near-full genome of ASFV and can be used for the genomic epidemiological analysis of ASFV outbreaks caused by different genotypes in any laboratory setting.

The authors stated that up to today, there are still gaps for WGS for ASFV, which has been challenging due to the size of the genome and the chemistries available. Therefore, they described a new approach to overcome these limitations, however they are some doubts on the number of samples used by the authors to evaluate this new methodology, along with no statistical analysis performed and only one genotype tested.

Please find below some comments respect to the Manuscript:

General comments:

In the manuscript the evaluation of this method is done only with 2 EDTA blood samples corresponding to pigs inoculated with Georgia 2007,  Genotype II or in silico. The number of samples is very limited, with only 2 EDTA blood pig samples evaluated, therefore increasing the number of samples, and including replicates will help to strengthen the results, as well as evaluation of different sample types, not only blood but tissues homogenates or swabs, to list some other sample types. Studies need to be expanded for a better understanding on performance of this new technology when different genotypes are to be evaluated. Consequently, it is not clear which will be the performance of the test when used to sequence different genotypes, or when recombinant viruses such as the GI/II described in China or Vietnam are in play.

It is of interest to expand into the Sensitivity of the test, and information such as the input DNA (expressed as by ng/ul, PCR Ct value, or titer) for the reaction to obtain a near complete WGS for ASF. Nanopore protocols, requires specific concentration for the different chemistries, it will be of great help to include which was the input used for these experiments. What happened when DNA input concentration is below or above the manufacturer recommendations? If this method is to be used in the field, are the samples to be normalized before setting up the reactions or any concentration as input can be used?

Material and Methods:

Lines 77-78: the authors refer another publication for the inoculation of the animals, however it will help to have some information listed on this manuscript, such as route of inoculation, days post inoculation that samples were taken, how many blood samples were collected and total number of pigs sampled. Any negative samples tested? Were the samples tested fresh, or samples were frozen and then evaluated?

Lines 88. Will be of interest if the authors can expand why Vaccinia virus was selected as surrogate for ASFV, and which are the similarities between both viruses that allow the data to be comparable.

Line 90. Swine testicles cells were used to grow the Vaccinia virus, please can the authors expand why blood was not used as matrix to spike the virus and mimic the sample type that will be taken in the field for ASFV, considering this is the sample of reference to detect ASFV?

Line 119: Is the alkaline buffer commercially available?

Line 153: The authors described the MinKNOW software use, but some additional information may be  beneficial to include, such as the q score for the nanopore runs.

Results:

Line 160: It is mentioned, the financial and practical constraints for ASF WGS. Then, it will be helpful if the authors can detail on which is the cost of the new method described in here vs other methods described, along with the comparison on time to implement and obtain data when this technology is use.

Lines 163 – Line 204: the authors present the concern on amplifying mitochondria or host chromosomal DNA; therefore, they used the SWGS package to specifically amplified ASF genome. Even when  targeting ASF, the mapped reads is above 20% when heat denaturation is used and over 70% when Alkaline treatment is performed. It will help for the interpretation, if  the authors can expand on the ratio of host vs mitochondrial vs virus reads, and why the ratio of mapped reads is still low when heat denaturation is implemented, even when using specific primers?

Line 176: How many replicates per temperature and incubation time were performed? A minimum set of 3 replicates along with standard deviation needs to be included to make the data more reliable and as a minimum to evaluate the consistency of the method.

Figure 1. Any explanation why it is a decreased of almost 3 Ct values / 1 log after 8 hours when samples are incubates at 42C.

How was the input material determined? Ct from input seems high, around 32. However, the authors didn’t evaluate samples with higher viral loads, such as Ct 20. Usually, a Ct of 20 is observed when ASF blood samples from animals showing clinical signs are tested. As previously mentioned, it will be of great interest to evaluate the test when different inputs are used and to include more information of the LOD of the test. The authors  recommendation for sample type used and inputs / cut off will be of great support for a better understanding on performance of this methodology.

Line 194: As previously mentioned, it is not clear on how many EDTA blood samples were tested. If only to, then running samples by triplicate will help on confidence of the results. Also, some comment on how sample quality can affect the test, will be beneficial.

Line 219: The authors indicated that” both approaches improve the efficiency of the specific amplification of ASF DNA”. The data shows over 73% mapped reads when alkaline denaturation vs 30% when heat denaturation was used. Please can the authors explain on how it is not a significantly difference between both denaturation protocols? This would be beneficial from some statistical analysis, as well.

How is significant an increase of 13% vs 20% using just two samples with not any replicates at least to consider some std dev?

Is this method suggested for rapid detection or characterization? If for characterization, which is the depth and coverage expected to be enough to call for specific indels or SNPs across the whole genome. What is the overall number of specific viral reads vs total number of reads expected?

Even when the method can reach 99% of coverage, the depth of coverage is very low (1X), therefore more samples should be included and improvements on the protocol need to be addressed to consider the method of selection over other approaches.

Line 247: authors stated that increase of overall depth may limit the number of samples on the ONT sequencing platform, however several publications using different viruses, have shown the capability of multiplexing several samples on ONT, when an amplicon approach is used, without having an impact on quality and still increasing the depth and maintaining capability to multiplex. The authors may want to expand on this statement.

The manuscript can be present as a proof of concept for development of a technique for primer design and implementation of MDA for ASF or DNA viruses, but the manuscript needs more work done in terms of number of samples, genotypes and replicates to make the data stronger.